# Veratridine-Induced Oscillations in Nav 1.7 but Not Nav 1.5 Sodium Channels Are Revealed by Membrane Potential Sensitive Dye

**DOI:** 10.3390/membranes15030080

**Published:** 2025-03-05

**Authors:** Sarah C. R. Lummis, Samantha C. Salvage, Christopher L.-H. Huang, Antony P. Jackson

**Affiliations:** 1Department of Biochemistry, University of Cambridge, Tennis Court Road, Cambridge CB2 1QW, UK; ss2148@cam.ac.uk (S.C.S.); clh11@cam.ac.uk (C.L.-H.H.); apj10@cam.ac.uk (A.P.J.); 2Physiology, Development and Neuroscience, University of Cambridge, Downing Place, Cambridge CB2 3DY, UK

**Keywords:** oscillations, depolarization, Flexstation, voltage-gated sodium channel, voltage-sensitive dye

## Abstract

Voltage-gated sodium channels (Navs) are critical for membrane potential depolarisation in cells, with especially important roles in neuronal and cardiomyocyte membranes. Their malfunction results in a range of disorders, and they are the target of many widely used drugs. A rapid yet accurate functional assay is therefore desirable both to probe for novel active compounds and to better understand the many different Nav isoforms. Here, we use fluorescence to monitor Nav function: cells expressing either the cardiac Nav 1.5 or pain-associated Nav 1.7 were loaded with fluorescent membrane potential sensitive dye and then stimulated with veratridine. Cells expressing Nav 1.5 show a concentration-dependent slow rise and then a plateau in fluorescence. In contrast, cells expressing Nav 1.7 show a more rapid rise and then unexpected oscillatory behavior. Inhibition by flecainide and mexiletine demonstrates that these oscillations are Nav-dependent. Thus, we show that this fluorescent membrane potential dye can provide useful functional data and that we can readily distinguish between these two Nav isoforms because of the behavior of cells expressing them when activated by veratridine. We consider these distinct behaviors may be due to different interactions of veratridine with the different Nav isoforms, although more studies are needed to understand the mechanism underlying the oscillations.

## 1. Introduction

Voltage-gated sodium channels (Navs) are vital for action potential generation and propagation in excitable cells such as cardiac myocytes. There are nine isoforms, Nav 1.1–Nav 1.9 [1,2,3,4]. Each Nav consists of a pore-forming α-subunit (Figure 1) and one or more regulatory β-subunits (β1–β4) [4,5,6]. Mutations in either subunit type can result in altered Nav function, leading to a range of conditions, some of which are potentially fatal. For example, mutations in cardiac Nav 1.5 are associated with Brugada syndrome, long QT syndrome type 3, sinus node disorder, atrial fibrillation, conduction defects, and sudden cardiac death [4,7]. Distinct mutations in Nav 1.7 are associated with a variety of inherited neuropathic pain syndromes including congenital insensitivity to pain and paroxysmal extreme pain disorder [8,9]. Nav channels are therefore a major target not only for research but also for pharmacological modulation [10]; some compounds that act on these channels are shown in Figure 1.

A range of methods have been used to study Navs of which the gold standard is patch clamp electrophysiology [11]. There has been considerable progress in the development of automated electrophysiological platforms, including higher throughput options [11], but they are expensive, and so attention has turned to other methods including the use of dyes to detect ion flux or changes in membrane potential. Of particular interest is the membrane potential sensitive dye produced by Molecular Devices Ltd., which uses a fluorescent indicator combined with a quencher. This performed well in tests to compare it with other membrane potential sensing dyes, including CC2 and DiSBAC2, and with a sodium sensing dye [12]. However, there were some unexpected results with some compounds that showed delayed onset, perhaps due to the use of veratridine to activate Navs and also some indication that some responses might oscillate, although this was not discussed. Veratridine-induced membrane potential oscillations have been observed since the 1970s, initially in muscle but also in non-excitable cells [12,13,14,15]. Given the significant increase in our knowledge in the Nav field in the last 50 years, we felt it was timely to explore the characteristics of veratridine-activated responses in two of the most interesting and widely studied Navs: Nav 1.5 and Nav 1.7.

## 2. Materials and Methods

Cell culture—Human embryonic kidney (HEK) 293 cells, and those stably expressing Nav 1.5 (HEK293-Nav 1.5 [16]) and Nav 1.7 (HEK293-Nav 1.7 [17]), were maintained on 90 mm tissue culture plates at 37 °C and 7% CO_2_ in a humidified atmosphere. They were cultured in Dulbecco’s Modified Eagle’s Medium/Nutrient Mix F12 (1:1) with GlutaMAX I media (Life Technologies, Paisley, UK) containing 10% HyClone fetal calf serum (GE Healthcare, Chicago, IL, USA). For Flexstation studies, cells were transferred to 96 well plates 2–4 days before assay.

FlexStation Studies—The methods were as described previously [17]. In brief, fluorescent membrane potential dye (Membrane Potential Blue kit, Molecular Devices, San Jose, CA, USA) was diluted in Flex buffer (10 mM HEPES, 115 mM NaCl, 1 mM KCl, 1 mM CaCl_2_, 1 mM MgCl_2_, and 10 mM glucose, pH 7.4) and added to each well. Following incubation at 37 °C for 30 min, fluorescence was measured in a FlexStation 3 (Molecular Devices) at 2 s intervals. Excitation and emission wavelengths were set to 485 nm and 525 nm, respectively, with a cutoff of 515 nm.

Veratridine (Abcam, Cambridge, UK) was prepared as a 10 mM stock solution in DMSO. Working concentrations were dissolved in Flex buffer and added to each well after 20 s. Inhibitors were also prepared as 10 mM stock solutions in DMSO and then diluted in Flex buffer. They were preincubated with the cells for at least 1 min before the assay.

Some data from single experiments (each with 3–4 wells) are shown with raw levels of fluorescence (F); to combine data from multiple experiments, data were normalized to the maximum change in fluorescence (Fmax, i.e., plateau value for Nav 1.5 and peak oscillation value for Nav 1.7) for that experiment. Data were analyzed using Prism (GraphPad Software Inc., San Diego, CA, USA)

Whole Cell Patch Clamp Electrophysiology—Whole cell recordings of Na^+^ currents were made on a four channel automated patch clamp system (Patchliner Quattro, Nanion Technologies, Munich, Germany). The extracellular solution for Nav 1.5 contained the following (in mM): NaCl 60, KCl 2, CaCl_2_ 1.5, glucose 10, MgCl_2_ 1, CsCl_2_ 90, and HEPES 10, pH 7.4 with NaOH; the Nav 1.7 recording solution was the same except it contained NaCl 140 mM and no CsCl_2_. This was in order to limit the driving force for Nav 1.5 currents, allowing for optimal voltage control. The intracellular solution contained (in mM) NaCl 35, CsF 105, EGTA 10, and HEPES 10, pH 7.2 with CsOH. Experiments were performed in the whole-cell configuration with HEKA EPC10 amplifiers using medium resistance (2–3 MΩ) NPC-16 chips (Nanion Technologies). Only cells with series resistances of 8 MΩ or less before 60–70% compensation were included, and leak currents were subtracted using a P/4 protocol [16,18]. All currents recorded were less than 5 nA, and data from cells with current amplitudes smaller than 100 pA prior to drug application were excluded. An initial current-voltage response was performed in normal extracellular solution for quality control, and then cells were incubated in each dose of flecainide or mexiletine for a period of 45 s–1 min. Any cells that did not complete the entire concentration–response range were excluded from analysis. The voltage protocol consisted of 100 ms test pulses ranging from −140 mV to +45 mV, in 5 mV increments, from a holding potential of −120 mv, 50 ms. Peak currents (*I*_Na_) were recorded from the −30 mV or −10 mV test pulse for Nav 1.5 and Nav 1.7, respectively. Data were visualized and analyzed in PatchMaster (v2x90.5) and IgorPro (v8.0.4.2) or Prism (GraphPad Prism v10.3.1).

## 3. Results

### 3.1. Responses to Veratridine

HEK293-Nav 1.5 cells loaded with membrane potential sensitive dye responded to veratridine in a concentration dependent manner (Figure 2a). Corresponding ‘wild-type’ HEK293 cells did not respond to veratridine, although they responded robustly to a depolarising stimulus of KCl (Figure 2b). HEK293-Nav 1.7 cells also responded to veratridine but unexpectedly showed oscillating behavior (Figure 2c). Concentration–response curves (Figure 3) revealed EC_50_s of 28 μM (pEC_50_ = 6.5 ± 0.04; data = mean ± SEM, *n*=6), and 8 μM (pEC_50_ = 5.1 ± 0.1; mean ± SEM, *n* = 6) for Nav 1.5 and Nav 1.7-expressing cells, respectively, consistent with previously published data [19,20,21].

### 3.2. Responses to Nav Inhibitors

The Nav inhibitor flecainide inhibited the veratridine-induced responses in a concentration-dependent manner. Typical data for inhibition of Nav 1.5-expressing cells in the presence of a range of concentrations of flecainide are shown in Figure 4a.

Multiple experiments revealed flecainide IC_50_s of 2 μM (pIC_50_ = 5.68 ± 0.09 M) and 13 μM (pIC_50_ = 4.86 ± 0.07 M), data = mean ± SEM, *n* = 4, for Nav 1.5 and Nav 1.7-expressing cells, respectively (Figure 5). The corresponding values for mexiletine were 9.3 μM (5.03 ± 0.09 M) and 30 μM (4.52 ± 0.08 M); data = mean ± SEM, *n* = 5–7. Typical data for the inhibition of Nav 1.7-expressing cells in the presence of a range of concentrations of mexiletine are shown in Figure 4b.

The effect of flecainide and mexiletine on Nav 1.5 and Nav 1.7 Na^+^ currents (*I*_Na_) were also explored under whole-cell voltage clamp conditions. Na^+^ currents from both Navs exhibited a concentration-dependent decrease in peak *I*_Na_ with flecainide or mexiletine, (typical responses are shown in Figure 6) with both compounds exhibiting a greater potency for Nav 1.5 compared to Nav 1.7: Flecainide inhibited peak *I*_Na_ with IC_50_ values of 8.9 µM (pIC_50_ = 5.04 ± 0.25 M) and 20.8 µM (pIC_50_ = 4.73 ± 0.08 M) for Nav 1.5 and Nav 1.7, respectively (Figure 6; data = mean ± SEM, *n* = 6). Mexiletine yielded IC_50_ values of 40.6 µM (pIC_50_ = 4.09 ± 0.88 M) and 437 µM (pIC_50_ = 3.50 ± 0.04) for Nav 1.5 and Nav 1.7, respectively (data = mean ± SEM, *n* = 6; Figure 6). These values are consistent with previously published data [22,23,24].

We also explored the effects of ouabain, a Na^+^/K^+^ pump inhibitor, and 4-amino pyridine (4-AP), a K^+^ channel inhibitor, on the veratridine-induced fluorescent responses. Ouabain at 1 mM (Figure 7) showed no significant inhibition of 30 μM veratridine-induced responses in Nav 1.5-expressing cells (responses were 87.6 ± 11% of control; data = mean ± SEM, *n* = 3) but did partially inhibit responses in Nav 1.7-expressing cells (responses were 54.6 ± 7% of control; data = mean ± SEM, *n* = 4). 4-AP had no significant inhibition on 30 μM veratridine-induced responses in Nav 1.7-expressing cells: responses were 120 ± 13% of control; data = mean ± SEM, *n* = 3. It was not tested in Nav 1.5-expressing cells.

## 4. Discussion

There are a number of ways to study Navs, but using membrane potential sensitive dyes has become increasingly popular recently, as it provides a relatively simple and rapid-throughput method of exploring function. It does require an activation mechanism and here we use veratridine, which has proved useful for this type of work (e.g., [12]). Our data reveal concentration-dependent increases in fluorescence in cells stably expressing either Nav 1.5 or Nav 1.7, with the latter showing oscillatory behavior. Biochemical oscillations are known to control a range of important aspects of cell physiology, including some involved in signaling and metabolism (see [25] for review), and so being aware of their existence is critical. Here, we show that different Nav isoforms display different responses to veratridine. These distinct responses, and the mechanism by which veratridine might elicit them, are discussed in more detail below.

Veratridine has been known for many decades to bind to voltage-dependent sodium channels and maintain them in an open state, thus inhibiting desensitization [26,27]. More recently, structures of veratridine bound to various Navs have been published and reveal more than one binding location in the pore, suggesting that veratridine may exhibit its effects via multiple binding sites, and these may differ subtly between the different Nav isoforms [20,28,29]. In our experiments, veratridine elicits a relatively slow increase in fluorescence (the half maximal response at 30 μM is reached in 1–2 min) in Nav 1.5-expressing cells. Experiments using the Nav antagonists flecainide and mexiletine inhibit these increases in fluorescence, demonstrating a dependence on Navs. These data suggest that veratridine binds to Navs that open spontaneously, and then keeps them in an open state, i.e., it acts in a similar manner to a classic agonist opening a ligand-gated ion channel. The responses we observe are slow compared to the millisecond activation in the patch clamp experiments, which is not surprising as the fluorescent response is from a population of cells and reflects the slow dye kinetics. The patch clamp experiments, where channels are opened using a voltage change, capture faster single-cell signals, so the responses have different characteristics.

The data show that in Nav 1.7 expressing cells veratridine activates channels more rapidly than those in Nav 1.5 expressing cells, but then they close and reopen multiple times generating oscillatory responses. Veratridine-generated membrane potential oscillations have been reported in the literature for more than 50 years [13,14,15], although it is only more recently that the roles of the different subfamilies of Nav channels have begun to be explored. It is now clear that different subfamilies are expressed in different tissues and generate different effects [2,4,30], although oscillatory characteristics associated with specific Navs have not—until now—been studied. Thus, it is not surprising that the characteristics of the published oscillations differed depending on the cell type, and—as these were primarily measured using electrophysiological techniques and not dyes—the responses obtained are a little different from ours. In myotubes for example, veratridine induces sudden depolarizations associated with bursts of action potentials, followed by repolarization and then hyperpolarization [31]. Oscillations have also been reported in non-electrically active cells, e.g., in glioma and neuroblastoma x glioma cells, where again, response characteristics differ both from each other and those in myotubes [12]. The oscillations we observed appear more regular than those observed using electrophysiological techniques, possibly because the movement of the voltage sensitive dye into and out of the membrane is slow relative to the very rapid opening and closing of Navs. It is also possible that other processes are involved. For example, veratridine modulates the Na^+^/K^+^ pump, and Samson and Brodies’ experiment {31} in myotubes suggests that this pump plays an important role in their veratridine-induced oscillatory behavior, although it is not yet clear if this is true for all preparations. The fact that ouabain inhibits responses in our Nav 1.7 cells indicates that the Na^+^/K^+^ pump may play a role in these cells, but more studies are required to confirm this. The activation of K^+^ channels has also been proposed to play a role, but our data showing no effect on Nav 1.7 responses by 4-AP suggest that K^+^ channels do not significantly contribute to these oscillations.

There is also the question of why in Nav 1.7 expressing cells—but not in Nav 1.5 expressing cells—veratridine results in channel opening followed by subsequent closure, and this cycle repeats multiple times. One plausible explanation is due to the voltage dependence of veratridine binding, i.e., it likely initially binds, but then unbinds when sufficient channels are open to increase the membrane potential to a level at which veratridine binding is unfavorable. Upon unbinding, the membrane potential would decrease, ultimately to a level where veratridine binding is again favored, and thus it would rebind and reopen the channels. If—as has been suggested—veratridine binds in different pore locations in Nav 1.5 and Nav 1.7 channels, it is likely to be differently influenced by the membrane potential, and thus would not respond to changes in membrane potential in the same manner. It is also possible that veratridine binds with different affinities in the different pores, which again would affect its binding and/or unbinding rates.

## 5. Conclusions

In conclusion, we have shown that membrane potential sensitive dye is suitable for studying and comparing different subfamilies of Nav channels and have revealed that veratridine activation results in oscillatory behavior in Nav 1.7-expressing cells, but not in Nav 1.5 expressing cells. Previous studies have also shown oscillations, including one using dyes; although, in the latter, no explanations for this behavior were given. We believe that more structural and functional studies are needed to clarify details of the mechanism that results in oscillations and also to determine how veratridine acts on the other Nav subfamilies.

## Figures and Tables

**Figure 1 membranes-15-00080-f001:**
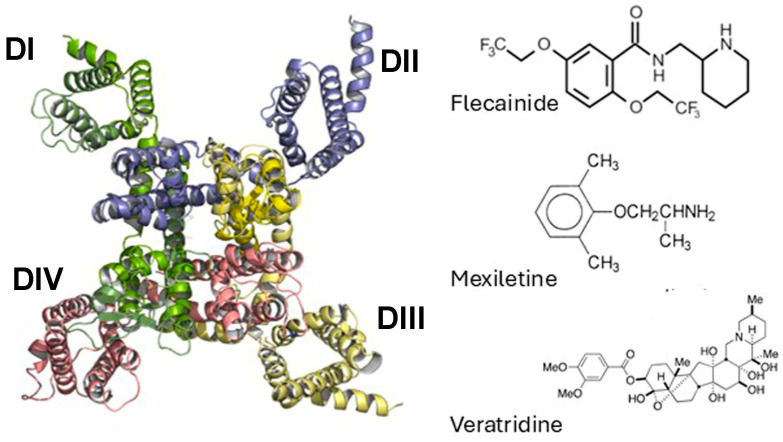
Top view structure of the human Nav 1.5 α subunit (PDB ID: 6LQA), with the 4 domains (DI–DIV) shown in different colors, alongside the compounds used in this study. All of these bind in the central cavity to modulate function.

**Figure 2 membranes-15-00080-f002:**
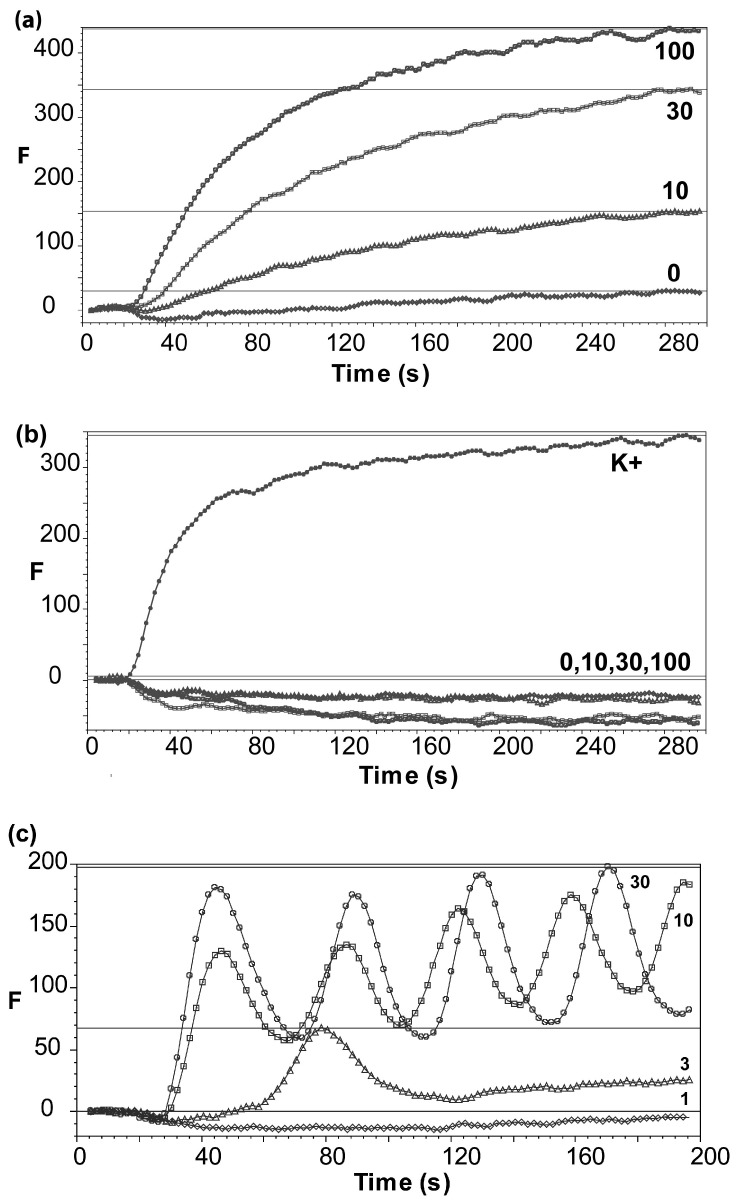
(**a**) Representative responses to varying veratridine concentrations (in μM) in cells expressing Nav 1.5; F = fluorescence, arbitrary units. Data from single wells. Veratridine was added at 20 s. (**b**) Application of the same concentrations of veratridine in untransfected cells showed no responses, although depolarization by K+ (100 mM KCl) yielded a robust response. (**c**) Representative responses to varying veratridine concentrations (in μM) in cells expressing Nav 1.7.

**Figure 3 membranes-15-00080-f003:**
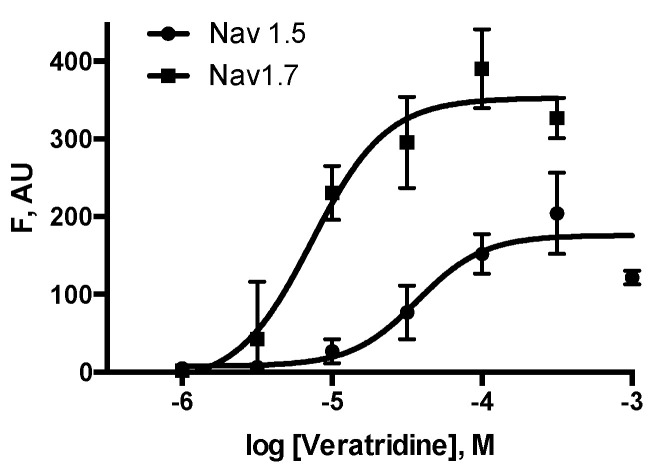
Veratridine concentration–response curves in cells expressing Nav 1.5 (circles) and Nav 1.7 (squares). F = fluorescence, arbitrary units. Data are mean ± SEM for 3–4 wells. Multiple experiments revealed an EC_50_s of 28 μM and 8 μM for veratridine activation of Nav 1.5-expressing cells and Nav 1.7-expressing cells, respectively.

**Figure 4 membranes-15-00080-f004:**
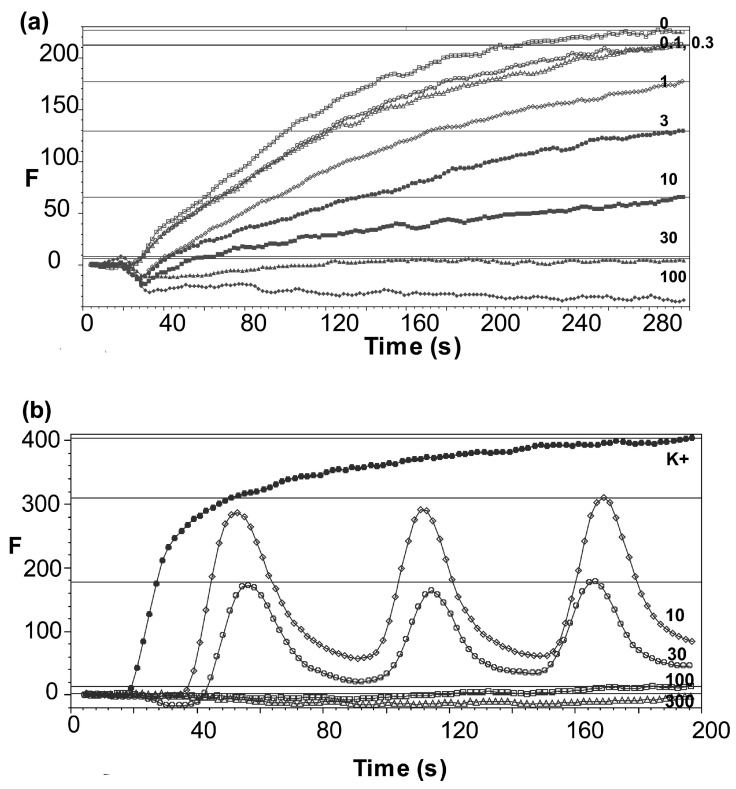
(**a**) Representative responses to veratridine (30 μM) in the presence of flecainide (concentrations as shown in μM) in cells expressing Nav 1.5; F = fluorescence, arbitrary units. Veratridine was added at 20 s. (**b**) Representative responses to veratridine (30 μM) in the presence of mexiletine (concentrations as shown in μM) in cells expressing Nav 1.7.

**Figure 5 membranes-15-00080-f005:**
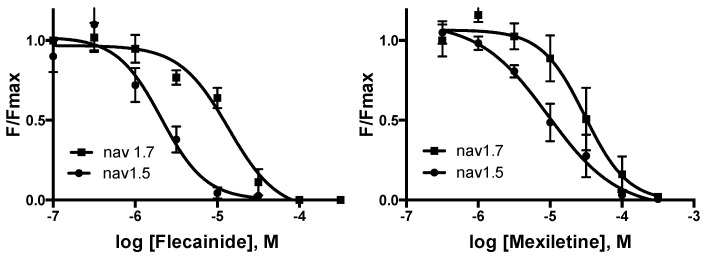
Concentration–response curves in cells expressing Nav 1.5 (circles) and Nav 1.7 (squares). Data = mean ± SEM, *n* = 5–7. Flecainide data revealed yielded IC_50_s of 2 μM and 13 μM, respectively, while those for mexiletine yielded values of 9 μM and 30 μM.

**Figure 6 membranes-15-00080-f006:**
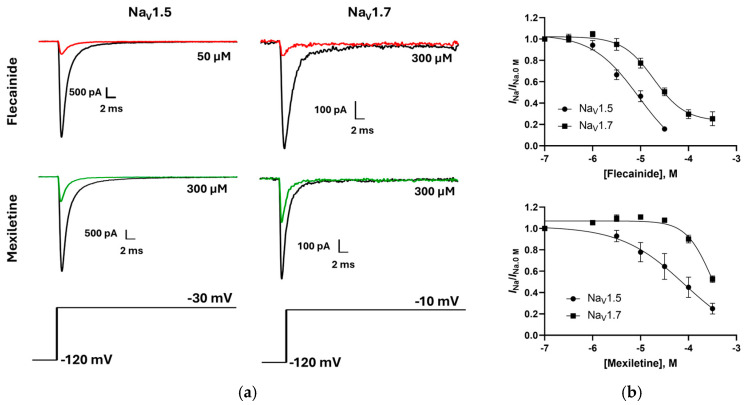
(**a**) Representative Nav 1.5 and Nav 1.7 currents pre (black) and post flecainide (red) or mexiletine (green) incubation. (**b**). Normalized concentration–response curves for flecainide (**upper panel**) and mexiletine (**lower panel**) on Nav 1.5 (circles) and Nav 1.7 (squares).

**Figure 7 membranes-15-00080-f007:**
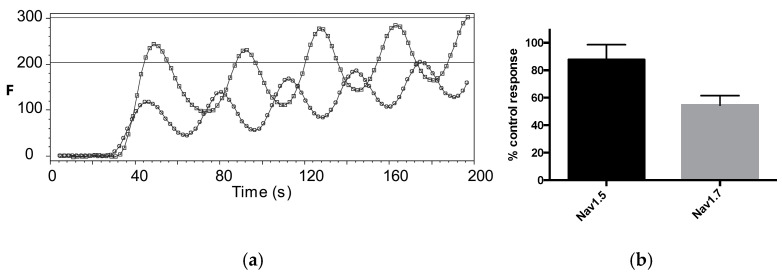
(**a**): Typical veratridine-induced fluorescent responses of Nav 1.7 expressing cells with (**lower trace**) and without (**upper trace**) 1 mM ouabain. Multiple experiments (**b**) show 54.6% inhibition at this concentration but no significant inhibitory effect at the same concentration in Nav 1.5-expressing cells. Data = mean ± SEM, *n* = 3–4.

## Data Availability

Data are contained within the article.

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
