# Peer review of "Veratridine-Induced Oscillations in Nav 1.7 but Not Nav 1.5 Sodium Channels Are Revealed by Membrane Potential Sensitive Dye"

_membranes, 2025, doi:10.3390/membranes15030080_

Round 1

Reviewer 1 Report

Comments and Suggestions for Authors

The manuscript titled “Veratridine-Induced Oscillations in Nav1.7 but Not Nav1.5 Sodium Channels Are Revealed by Membrane Potential-Sensitive Dye” by Sarah C. R. Lummis, Samantha C. Salvage, Christopher L-H. Huang, and Antony P. Jackson presents novel findings about unique veratridine-induced oscillations in Nav1.7 sodium channels, measured using a membrane potential-sensitive dye. The oscillations induced by veratridine occurred in cells expressing Nav1.7 but not Nav1.5.

The manuscript contains novel findings that are a significant contribution to the field. However, the presentation has several errors and inconsistencies, making the manuscript difficult to follow. The selective action of veratridine-induced oscillations is an intriguing observation, but the clarity and accuracy of the data need improvement. I have identified the following points that need to be addressed, though there may be additional issues that require attention:

  1. Page 2, line 66: The reference number for Price and Lummis, 2005 is missing.
  2. Page 2, Materials and Methods: Lines 60-61 state “cells … were maintained on 90 mm … plates …”, but lines 67-70 mention “… dye … added to each well”. Were the cells transferred to a 96-well plate?
  3. Fluorescent Measurements: Please provide more details on the fluorescent measurements of membrane potential dye, including the excitation and emission wavelengths.
  4. Page 2, line 75: Clarify the sentence: “Data were normalized to the maximum ∆F …”.
  5. Page 3, Line 87: The P/4 protocol needs further explanation or a citation (e.g., Armstrong CM, Bezanilla F. 1974; DOI: 10.1085/jgp.63.5.533).
  6. Page 3, lines 89-91: This section repeats information already stated in Page 2, Line 73 ("Inhibitors were prepared as 10 mM stock solutions in DMSO …").
  7. Page 3, lines 97-98: Please specify the version numbers for PatchMaster and Igor Pro software.
  8. Figures 1 and 4: The axis labels on these graphs are too small to read. 
  9. Figures 1 and 4: The y-axis label states "F" in arbitrary units, but the Materials and Methods section indicates "Data were normalized to the maximum ∆F …," which appears contradictory.
  10. Figure 2: Verify if the labeling of "Nav1.5 (circles)" and "Nav1.7 (squares)" is accurate. The representative traces in Figure 1 suggest Nav1.5 reaches ~500 AU, while Nav1.7 is below 200 AU.
  11. Figure 2: Does the data represent the plateau value for Nav1.5 and the peak oscillatory response for Nav1.7? This should be explained.
  12. Figure 2 Legend: The statement "Veratridine was added at 20s" is unnecessary here (and also applies to Figure 3).
  13. Figure 2 Legend: Clarify what "n=4" represents—data from four wells or four trials?
  14. Figure 2 Legend: The phrase "These and similar data revealed …" is unclear; please rephrase.
  15. Page 3, lines 107-108: Correct the inconsistency regarding "n" values. The text states n=6, but the Figure 2 legend states n=4.
  16. Figure 3: There is no explanation for the error bars.
  17. Figure 4: Introduce this figure before Figure 3 for better flow..
  18. Page 5, line 211: The statement "a greater potency for Nav1.5" is unclear in the context of Figure 5, possibly due to the high concentration of Flecainide.
  19. Page 7, lines 258-265: Specify whether this paragraph discusses the effects of inhibitors on fluorescence signals or currents.
  20. Page 9, line 311: Use consistent terminology—replace "voltage-sensitive dye" with "membrane potential-sensitive dye."
  21. Discussion Section: Expand on the differences between optical membrane potential recordings and whole-cell patch clamp recordings. Highlight that optical signals represent slower population responses, while patch clamp data capture faster, single-cell signals. The slow dye kinetics likely influence the interpretation of oscillations observed in this study. 

Author Response

The manuscript contains novel findings that are a significant contribution to the field. However, the presentation has several errors and inconsistencies, making the manuscript difficult to follow. The selective action of veratridine-induced oscillations is an intriguing observation, but the clarity and accuracy of the data need improvement. I have identified the following points that need to be addressed, though there may be additional issues that require attention:

  1. Page 2, line 66: The reference number for Price and Lummis, 2005 is missing.

Now added

  1. Page 2, Materials and Methods: Lines 60-61 state “cells … were maintained on 90 mm … plates …”, but lines 67-70 mention “… dye … added to each well”. Were the cells transferred to a 96-well plate?

The cells were transferred. This has now been added to the methods section

  1. Fluorescent Measurements: Please provide more details on the fluorescent measurements of membrane potential dye, including the excitation and emission wavelengths.

These details have now been added

  1. Page 2, line 75: Clarify the sentence: “Data were normalized to the maximum ∆F …”.

This has been expanded to clarify

  1. Page 3, Line 87: The P/4 protocol needs further explanation or a citation (e.g., Armstrong CM, Bezanilla F. 1974; DOI: 10.1085/jgp.63.5.533).

These details have been added

  1. Page 3, lines 89-91: This section repeats information already stated in Page 2, Line 73 ("Inhibitors were prepared as 10 mM stock solutions in DMSO …").

This has been removed

  1. Page 3, lines 97-98: Please specify the version numbers for PatchMaster and Igor Pro software.

These details have been added

  1. Figures 1 and 4: The axis labels on these graphs are too small to read. 

These have been enlarged

  1. Figures 1 and 4: The y-axis label states "F" in arbitrary units, but the Materials and Methods section indicates "Data were normalized to the maximum ∆F …," which appears contradictory.

Raw data are shown with F, combined data for analysis were normalised. These details have been added to the methods.

  1. Figure 2: Verify if the labeling of "Nav1.5 (circles)" and "Nav1.7 (squares)" is accurate. The representative traces in Figure 1 suggest Nav1.5 reaches ~500 AU, while Nav1.7 is below 200 AU.

Yes, maximum responses depend on the number of cells present which varies from experiment to experiment

  1. Figure 2: Does the data represent the plateau value for Nav1.5 and the peak oscillatory response for Nav1.7? This should be explained.

Yes these details have been added

  1. Figure 2 Legend: The statement "Veratridine was added at 20s" is unnecessary here (and also applies to Figure 3).

Now removed

  1. Figure 2 Legend: Clarify what "n=4" represents—data from four wells or four trials?

Now clarified

  1. Figure 2 Legend: The phrase "These and similar data revealed …" is unclear; please rephrase.

Now rephrased

  1. Page 3, lines 107-108: Correct the inconsistency regarding "n" values. The text states n=6, but the Figure 2 legend states n=4.

Now corrected

  1. Figure 3: There is no explanation for the error bars.

Now explained

  1. Figure 4: Introduce this figure before Figure 3 for better flow..

Now moved

  1. Page 5, line 211: The statement "a greater potency for Nav1.5" is unclear in the context of Figure 5, possibly due to the high concentration of Flecainide.

We agree this is unclear and have now revised this sentence

  1. Page 7, lines 258-265: Specify whether this paragraph discusses the effects of inhibitors on fluorescence signals or currents.

Now clarified

  1. Page 9, line 311: Use consistent terminology—replace "voltage-sensitive dye" with "membrane potential-sensitive dye."

Now corrected

  1. Discussion Section: Expand on the differences between optical membrane potential recordings and whole-cell patch clamp recordings. Highlight that optical signals represent slower population responses, while patch clamp data capture faster, single-cell signals. The slow dye kinetics likely influence the interpretation of oscillations observed in this study. 

We have expanded this part of the discussion

Reviewer 2 Report

Comments and Suggestions for Authors

Thanks to a membrane potential fluorescent probe, the authors shown that both stimulation of Nav1.5 and Nav1.7 channels by veratridine induce a depolarization (in HEK cells transfected with these channels). For the Nav1.5 channel, the depolarization was stable, while for the Nav1.7 channel, it induced oscillating depolarisations. Sodium channel antagonists (flecainide and mexiletine) inhibited the depolarisations (stable or oscillating). These inhibitors inhibited the activity of sodium channels (measured using the patch-clamp technique) at similar concentrations. The results on the membrane potential probe are not very new; on the other hand, those on the oscillations are original.

Main point: 

Oscillations : The oscillations should be better investigated. Are these oscillations those of a single cell or a population of cells? Can these oscillations be measured thanks to the patch-clamp technique, and are they similar to those measured by fluorescence? With the patch-clamp, voltage clamp mode, does a stimulation mimicking the oscillations of potentials induce a sodium current? Does the window current of these channels participate in these oscillations?

Minor point:

 Since the results regarding ouabain are discussed, it would be interesting to show the results.

Author Response

Thanks to a membrane potential fluorescent probe, the authors shown that both stimulation of Nav1.5 and Nav1.7 channels by veratridine induce a depolarization (in HEK cells transfected with these channels). For the Nav1.5 channel, the depolarization was stable, while for the Nav1.7 channel, it induced oscillating depolarisations. Sodium channel antagonists (flecainide and mexiletine) inhibited the depolarisations (stable or oscillating). These inhibitors inhibited the activity of sodium channels (measured using the patch-clamp technique) at similar concentrations. The results on the membrane potential probe are not very new; on the other hand, those on the oscillations are original.

Main point: 

Oscillations : The oscillations should be better investigated. We agree, but, as described in the discussion,  more detailed work is beyond the scope of this paper.

Are these oscillations those of a single cell or a population of cells? The Flexstation measures a population of cells; we have now clarified this in  the methods.

Can these oscillations be measured thanks to the patch-clamp technique, and are they similar to those measured by fluorescence?

We have not observed any oscillations in the patch-clamp studies, although given that the voltage is clamped and depolarisation steps are in the order of ms, we would not expect to observe the oscillations seen in the fluorescence studies. Others that have seen veratridine induced oscillations in cells – and which are described in the discussion - have used other electrophysiological methods.

With the patch-clamp, voltage clamp mode, does a stimulation mimicking the oscillations of potentials induce a sodium current?

This is an interesting idea but we do not have the technical equipment to perform such an experiment. One experiment we theoretically could do would be to clamp the potential and then depolarise to a separate potential, but that would not reflect the dynamic variation of current observed in the oscillations.  Also, as the oscillations we observed are initiated by veratridine, and follow such a different time course to those in the fluorescence experiments, it is unclear how informative such an experiment would be.

Does the window current of these channels participate in these oscillations?

We consider this is extremely unlikely given the small size of the window current (2-5% of peak current at -40—50mV in Nav1.7 Ref16). Also non-excitable cells such as HEKs have lower resting membrane potentials (RMP: -20 to -30 mV) which would sit outside this window current. In addition veratridine causes a significant hyperpolarisation of the steady-state inactivation curve (Ref 25 at 7 uM  -20 mV shift and Ref 18 at 75 uM; -10 mV shift), with little effect on the conductance/activation curve at lower doses (Ref 25) and a small hyperpolarising shift at higher doses (Ref 18; ~6.5 mV). Collectively, these effects would be anticipated to further reduce the magnitude of any window current and shift it to even more hyperpolarised potentials relative to the RMP of the HEK cells.

.

Minor point:

 Since the results regarding ouabain are discussed, it would be interesting to show the results.

We have added these data ( new figure 6)

Reviewer 3 Report

Comments and Suggestions for Authors

The article “Veratridine- induced oscillations in Nav 1.7 but not Nav 1.5 sodium channels are revealed by membrane potential sensitive dye” is intended propose the use a of fluorescent membrane potential sensitive dye, in order to detect the activity of Na+ channels under different stimuli; however, some important issues should be pointed out:

1.- Please add on Figure 1 a 3D protein model of the Nav 1.7 and Nav 1.5 sodium channels, and the structures of the drugs, to a better understanding of the interaction.

2.- There is more fluorescent dyes to evaluate membrane potential, as DiBaC4, why your group or work do not perform a direct comparison with this well-known dye?

3.- Please add the conditions of the detection of the membrane 67 potential blue kit (wavelength of excitation and emission), there is not a clear description in M&M section 

4.- Line 258. Add the charges to Na+ and K+

Author Response

The article “Veratridine- induced oscillations in Nav 1.7 but not Nav 1.5 sodium channels are revealed by membrane potential sensitive dye” is intended propose the use a of fluorescent membrane potential sensitive dye, in order to detect the activity of Na+ channels under different stimuli; however, some important issues should be pointed out:

1.- Please add on Figure 1 a 3D protein model of the Nav 1.7 and Nav 1.5 sodium channels, and the structures of the drugs, to a better understanding of the interaction.

We have added a model of a Nav and the structures of the drugs (new figure 1) 

2.- There is more fluorescent dyes to evaluate membrane potential, as DiBaC4, why your group or work do not perform a direct comparison with this well-known dye?

This is an interesting idea, but we felt it was unnecessary especially as there is a previous study comparing FMP and DiBacC4 (Finley, 2001) where the former was preferred.  This has now been mentioned in the introduction

3.- Please add the conditions of the detection of the membrane 67 potential blue kit (wavelength of excitation and emission), there is not a clear description in M&M section

This has now been added

4.- Line 258. Add the charges to Na+ and K+

Now added

Round 2

Reviewer 3 Report

Comments and Suggestions for Authors

No further comments